# Catalytic asymmetric Tsuji–Trost α−benzylation reaction of N-unprotected amino acids and benzyl alcohol derivatives

Jian-Hua Liu [1], Wei Wen [1 ✉], Jian Liao[1], Qi-Wen Shen[1], Yao Lin[1], Zhu-Lian Wu [1], Tian Cai [1] & Qi-Xiang Guo [1 ✉]

Catalytic asymmetric Tsuji–Trost benzylation is a promising strategy for the preparation of chiral benzylic compounds. However, only a few such transformations with both good yields and enantioselectivities have been achieved since this reaction was first reported in 1992, and its use in current organic synthesis is restricted. In this work, we use N-unprotected amino acid esters as nucleophiles in reactions with benzyl alcohol derivatives. A ternary catalyst comprising a chiral aldehyde, a palladium species, and a Lewis acid is used to promote the reaction. Both mono- and polycyclic benzyl alcohols are excellent benzylation reagents. Various unnatural optically active α-benzyl amino acids are produced in good-to-excellent yields and with good-to-excellent enantioselectivities. This catalytic asymmetric method is used for the formal synthesis of two somatostatin mimetics and the proposed structure of natural product hypoestestatin 1. A mechanism that plausibly explains the stereoselective control is proposed.

---

[1] Key Laboratory of Applied Chemistry of Chongqing Municipality, and Chongqing Key Laboratory of Soft-Matter Material Chemistry and Function Manufacturing, School of Chemistry and Chemical Engineering, Southwest University, 400715 Chongqing, China. ✉email: wenwei1989@swu.edu.cn; qxguo@swu.edu.cn

The catalytic asymmetric Tsuji–Trost allylation[1–9] and benzylation[10–12] reactions are important strategies for enantioselectively constructing carbon–carbon and carbon–heteroatom bonds. Since it was first reported in 1970[13,14], catalytic asymmetric allylation has been studied extensively and now has widespread applications in organic synthesis[15–18]. However, few studies of catalytic asymmetric benzylation reactions have been reported[19–26]. This is probably because Tsuji–Trost allylation proceeds via a stable $\eta^3$-allyl–palladium intermediate, whereas benzylation takes place via an unstable aromaticity-disrupted $\eta^3$-benzyl–palladium complex (Fig. 1a)[10–12]. Catalytic asymmetric benzylation is therefore more challenging. Especially, the successful transformations of the catalytic asymmetric benzylation of prochiral nucleophiles were very limited. In 2010, Trost and Czabaniuk reported the palladium-catalyzed asymmetric benzylation of 3-aryloxindoles to chiral 3,3'-disubstituted oxindoles in excellent yields and with excellent enantioselectivities[27]. Subsequently, the catalytic asymmetric benzylation of azlactones was reported by the same research group[28,29]. In 2016, Tabuchi et al. found that the catalytic asymmetric benzylation of active methylene compounds proceeded via a dynamic kinetic asymmetric transformation[30]. In 2018, Snaddon and coworkers used a combined catalytic system derived from a chiral Lewis base and a palladium species for the catalytic asymmetric benzylation of α-aryl- or α-alkenyl-acetic acid esters[31]. Recently, two highly efficient catalytic asymmetric benzylation reactions of sec-phosphine oxides have been reported by Zhang and Liu[32], and Zhang[33], respectively (Fig. 1b). The development of new Tsuji–Trost-type benzylation reactions is therefore important because this will provide new methods for the preparation of optically active benzylic compounds and enable new applications of this named reaction in organic synthesis.

The direct catalytic asymmetric α−benzylation of N-unprotected amino acids is one of the simplest methods for preparing optically active unnatural α−benzyl amino acids. However, the strong nucleophilicity of the amino group and the weak α−carbon acidity of the N-unprotected amino acid make this direct benzylation challenging. Until now, this type of prochiral nucleophile has not been used in Tsuji–Trost benzylation reactions (Fig. 1b)[34,35]. The chiral aldehyde catalysis based on imine activation disclosed by our group provides a promising strategy for achieving this challenging reaction because the chiral aldehyde catalyst can simultaneously mask the amino group and enhance the α−carbon acidity via in situ formation of an imine with a N-unprotected amino acid ester group[36–43].

Here, we report a direct catalytic asymmetric Tsuji–Trost benzylation of N-unprotected amino acids. Use of a ternary catalytic system comprising a chiral aldehyde, a palladium species, and a Lewis acid enables the formation of the corresponding optically active α−benzyl amino acids in good-to-excellent yields and with good-to-excellent enantioselectivities (Fig. 1c). The use of this reaction for the synthesis of structurally diverse unnatural amino acids, somatostatin mimetics, and the natural product hypoestestatin 1, and identification of a possible reaction mechanism, are explored.

## Results

**Optimization of reaction conditions.** Our work began with the evaluation of the benzylation of ethyl alaninate (**1a**) with *tert*-butyl (naphthalen-2-ylmethyl) carbonate (**2a**) by using a combined catalytic

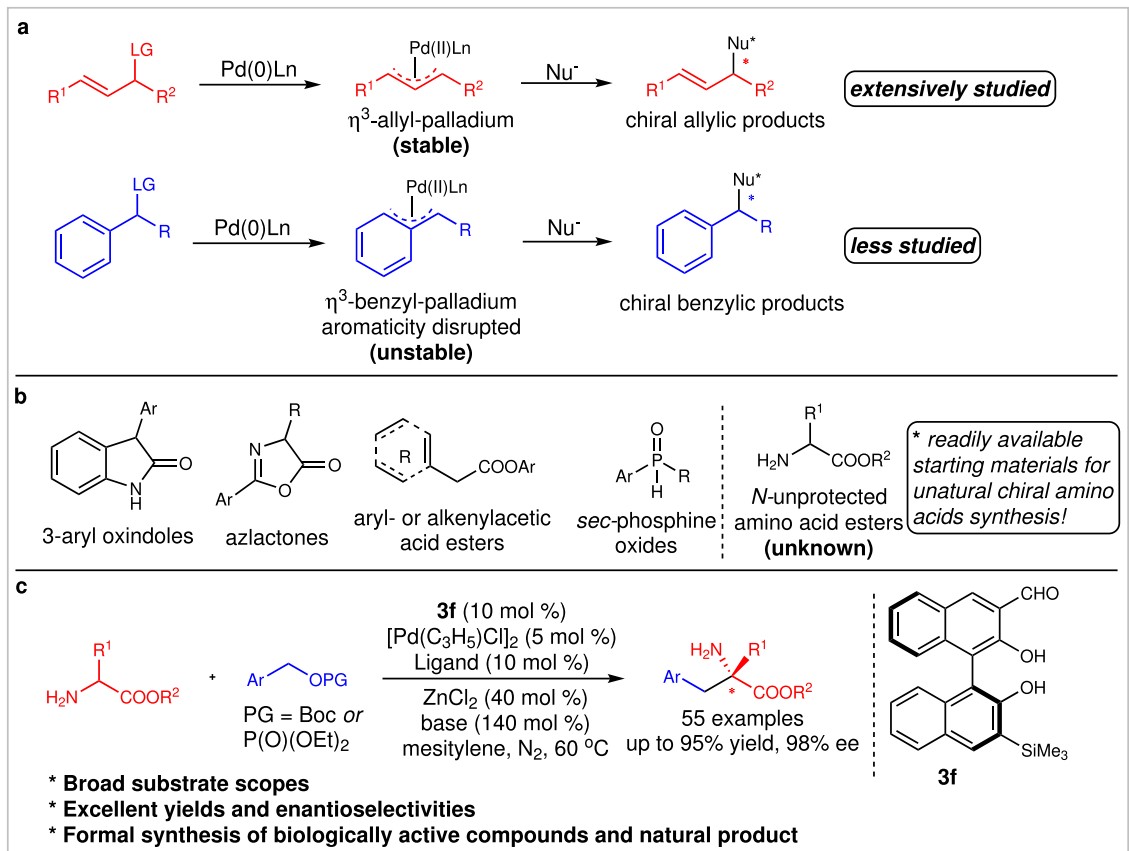

**Fig. 1 The catalytic asymmetric Tsuji–Trost-type reactions. a** The comparison of the allylation and benzylation reactions. **b** The reported prochiral nucleophiles involved in the catalytic asymmetric benzylation reactions. **c** The direct catalytic asymmetric α−benzylation of N-unprotected amino acid esters by a chiral aldehyde-involved catalytic system (this work).

system[44–46] consisting of chiral aldehyde **3h** (10 mol%), a palladium complex (5 mol%), and zinc chloride (40 mol%) in toluene, with the assistance of 1,3-bis(diphenylphosphanyl)propane (dppp, 10 mol%) as a ligand and the base 1,1,3,3-tetramethylguanidine (TMG, 100 mol %). This reaction proceeded smoothly at 60 °C in an inert atmosphere to give the desired product **5a** in 51% yield and with 76% enantioselective excess (ee) (Table 1, entry 1). These conditions were adopted as the standard conditions and optimization of these conditions was investigated systematically. First, various chiral aldehydes were used as catalysts instead of **3h** (Table 1, entry 2 and Supplementary Table 1). Among catalysts **3** and **4**, chiral aldehyde **3m** showed the highest catalytic activation ability in this reaction, and product **5a** was obtained in the highest yield (85%) in 4hs. However, the enantioselectivity was lower than that achieved with **3h** (62% ee vs 76% ee). Chiral aldehyde **4c** gave **5a** with the highest enantioselectivity (78% ee), but the yield was low (6%). Next, we tested various palladium sources in this reaction. We found that this reaction is sensitive to the palladium source; [Pd(C₃H₅)Cl]₂ gave the best results (Table 1, entry 3 and Supplementary Table 2). The ligands, Lewis acids, and bases were then sequentially screened, but no better results were obtained (Table 1, entries 4–6; Supplementary Tables 3–5). Screening of the alkoxy groups of amino esters **1** and leaving groups of benzyl alcohols **2** did not lead to further improvements in the yield and enantioselectivity of **5a** (Table 1, entries 7 and 8; Supplementary Tables 6 and 7). Various solvents were then screened; the use of mesitylene improved the yield to 94%, but the enantioselectivity decreased slightly (Table 1, entry 9 and Supplementary Table 8). Mesitylene was used as the solvent in the further screening of the bases and chiral aldehyde catalysts. We found that the combination of super

organic base tris(dimethylamino)iminophosphorane (TDMAIP) (1.4 equivalents) and chiral aldehyde catalyst **3f** could give **5a** in 94% yield and with 84% ee (Table 1, entry 10). The enantioselectivity of **5a** was further enhanced by doubling the reactant concentrations (Table 1, entry 11). On the basis of these results, the reaction conditions in Table 1, entry 12 were identified as the optimal conditions and were used in subsequent investigations.

**Substrate scope with amino acids.** With the optimal reaction conditions in hand, we then investigated the amino acid and arylmethanol substrate scopes. First, various amino acids bearing linear α−alkyl groups were introduced as reactants. The results indicated that amino acids containing linear alkyls with one–six carbon atoms all reacted efficiently with **2a** to give products **5a–5f** in excellent yields and with excellent enantioselectivities. The reactivities of amino acids with α-branched alkyls are greatly affected by the steric effects. For example, the ethyl valinate could not participate in this reaction under the optimal reaction conditions (Fig. 2, **5g**), whereas the ethyl 2-amino-2-cyclopropylacetate and ethyl leucinate gave corresponding products in good yields and enantioselectivities (Fig. 2, **5h–5i**). The use in this reaction of amino acid esters bearing phenyl, C=C, amino, or sulfur ether groups on their side chains was then examined. All these amino acid esters reacted efficiently with **2a** to give **5j–5o** in good-to-excellent yields and enantioselectivities. α-phenyl glycine ethyl ester was also a good reaction partner for **2a**, giving product **5p** in 86% yield and 78% ee. When diethyl glutamate was used as the donor in the reaction with **2a**, a

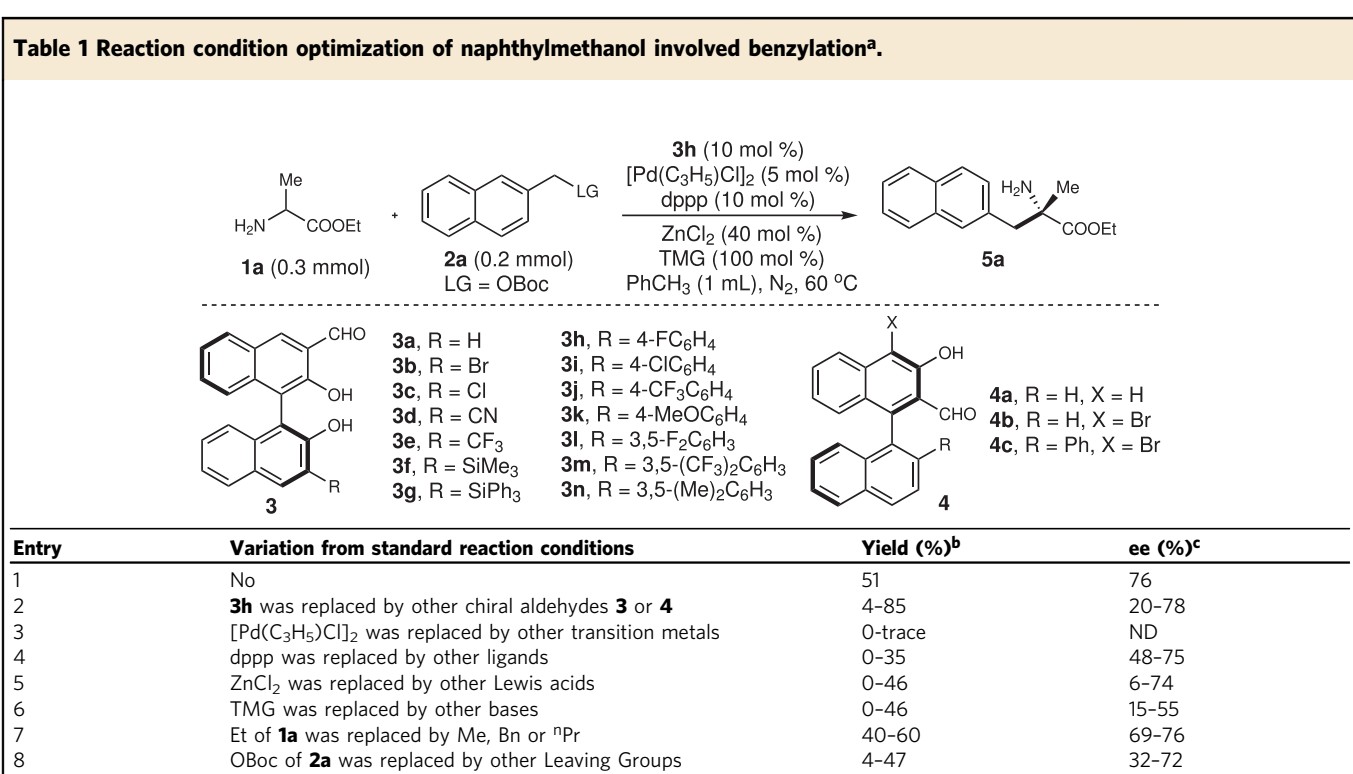

**Table 1 Reaction condition optimization of naphthylmethanol involved benzylation[a].**

| Entry | Variation from standard reaction conditions | Yield (%)[b] | ee (%)[c] |
|---|---|---|---|
| 1 | No | 51 | 76 |
| 2 | **3h** was replaced by other chiral aldehydes **3** or **4** | 4–85 | 20–78 |
| 3 | [Pd(C₃H₅)Cl]₂ was replaced by other transition metals | 0-trace | ND |
| 4 | dppp was replaced by other ligands | 0–35 | 48–75 |
| 5 | ZnCl₂ was replaced by other Lewis acids | 0–46 | 6–74 |
| 6 | TMG was replaced by other bases | 0–46 | 15–55 |
| 7 | Et of **1a** was replaced by Me, Bn or ⁿPr | 40–60 | 69–76 |
| 8 | OBoc of **2a** was replaced by other Leaving Groups | 4–47 | 32–72 |
| 9 | PhCH₃ was replaced by mesitylene | 95 | 74 |
| 10 | mesitylene as solvent, TDMAIP as base, and **3f** as catalyst | 94 | 84 |
| 11 | entry 10 and 0.5 mL mesitylene as solvent | 93 | 90 |

*ND* not determined.
[a]Reaction conditions: **1** (0.30 mmol), **2** (0.20 mmol), **3** (0.02 mmol), dppp (0.02 mmol), [Pd(C₃H₅)Cl]₂ (0.01 mmol), TMG (0.20 mmol), and ZnCl₂ (0.08 mmol) were stirred in toluene (1.0 mL) at 60 °C for indicated time.
[b]Isolated yield.
[c]The enantioselective excess was determined by chiral HPLC.

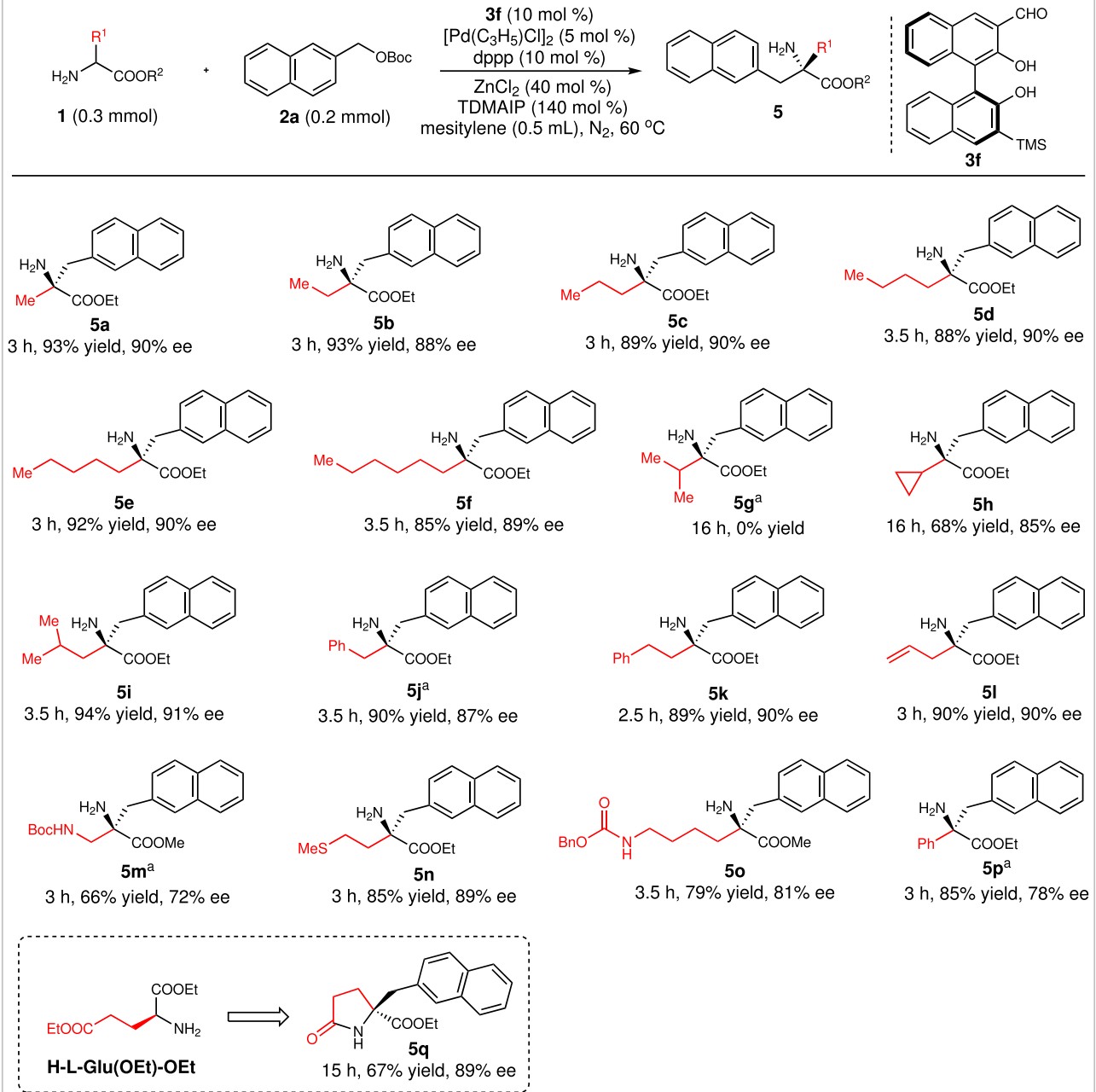

**Fig. 2 Substrate scope of amino acids.** Reaction conditions: **1** (0.30 mmol), **2a** (0.20 mmol), **3f** (0.02 mmol), dppp (0.02 mmol), [Pd(C$_3$H$_5$)Cl]$_2$ (0.01 mmol), tris(dimethylamino)iminophosphorane (TDMAIP, 0.20 mmol), and ZnCl$_2$ (0.08 mmol) were stirred in mesitylene (0.5 mL) at 60 °C under the nitrogen atmosphere for indicated time. Isolated yields. The enantioselective excess (ee) was determined by HPLC. [a]At 80 °C.

tandem benzylation–lactamization proceeded smoothly to give γ-lactam **5q** in 67% yield and with 89% ee. This type of chiral γ-lactams was extensively used as building blocks in the asymmetric synthesis of natural products[47–50].

**Substrate scope with polycyclic arymethyl alcohol derivatives.** The polycyclic arylmethyl alcohol substrate scope was investigated. 2-Naphthylmethanol derivatives bearing electron-donating or electron-withdrawing groups were good reaction partners for amino acid ester **1a**, and gave products **6a**–**6c** in excellent yields and enantioselectivities. 1-Naphthylmethanol derivatives also reacted with **1a** efficiently, but the required reaction temperature was higher than that for the reactions of 2-naphthylmethanols

and **1a** (Fig. 3, **6d**–**6h**). This may be because of the steric effect of the 1-naphthyl group. A series of nitrogen-containing aryl-methanol derivatives were then tested. All these compounds reacted efficiently with **1a** to give the corresponding products **6i**–**6n** in excellent experimental outcomes. Tricyclic arylmethanol derivatives, such as anthracen-2-ylmethanol and phenanthren-9-ylmethanol derived *tert*-butyl carbonates gave products **6o**–**6q** in excellent yields and enantioselectivities. Adapalene is a drug that is used in the treatment of skin diseases. Modification of this drug molecule was achieved by the catalytic asymmetric α-benzylation of amino acid ester **1a** with the adapalene-derived *tert*-butyl carbonate; **6r** was obtained in 95% yield and with 88% ee. We found the monocyclic *tert*-butyl benzyl carbonate could not react with **1a** under the optimal reaction conditions.

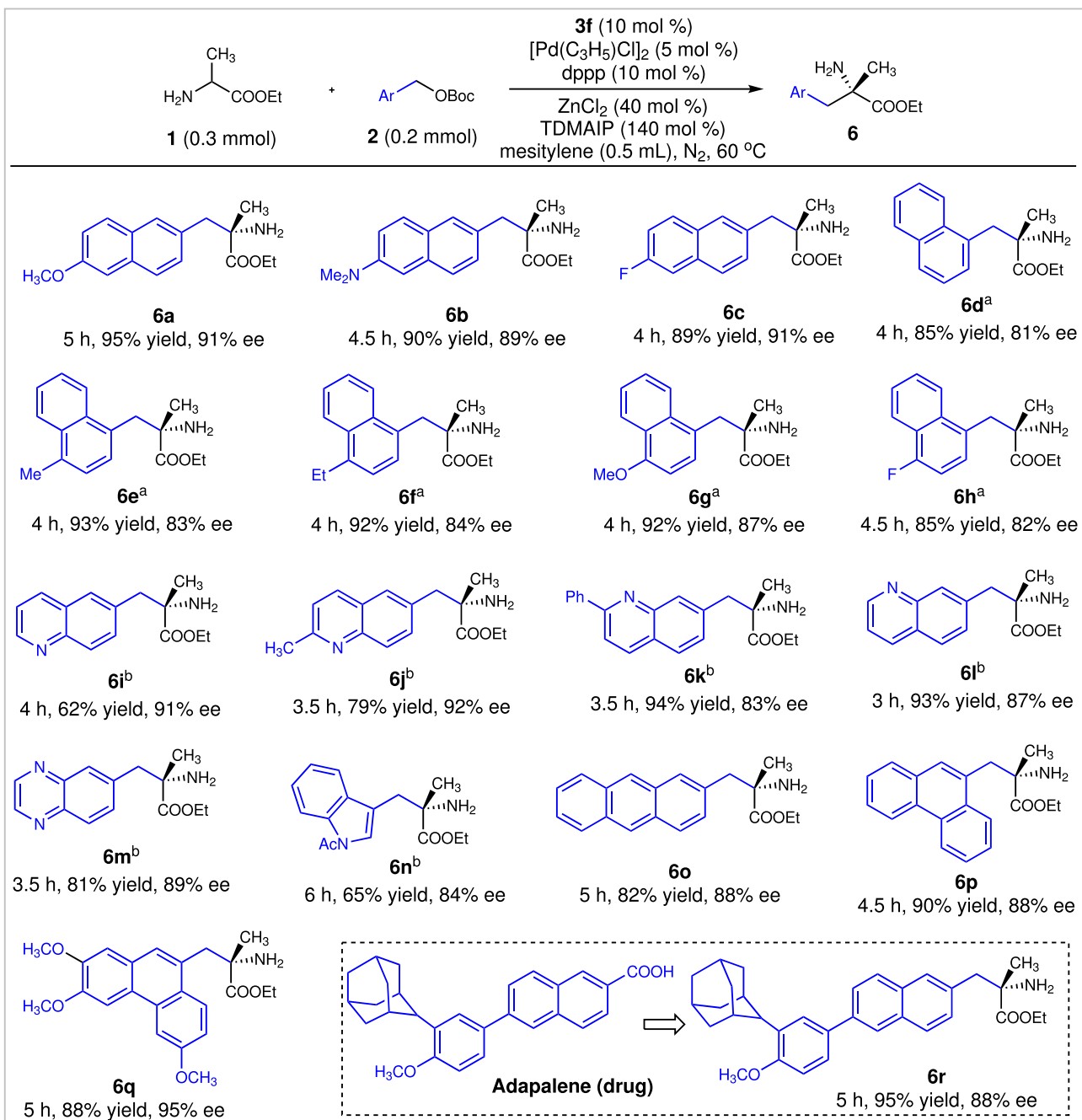

**Fig. 3 Substrate scope of polycyclic benzyl alcohol derivatives.** Reaction conditions: **1a** (0.30 mmol), **2** (0.20 mmol), **3f** (0.02 mmol), dppp (0.02 mmol), [Pd(C₃H₅)Cl]₂ (0.01 mmol), tris(dimethylamino)iminophosphorane (TDMAIP, 0.20 mmol), and ZnCl₂ (0.08 mmol) were stired in mesitylene (0.5 mL) at 60 °C under the nitrogen atmosphere for indicated time. Isolated yields. The enantioselective excess (ee) was determined by HPLC. [a]At 80 °C. [b]Yields and ees were given by the N-Boc-protected derivatives.

**Optimization of the monocyclic benzyl alcohol involved benzylation reaction**. To further expand the substrate scope, the reaction conditions of the asymmetric benzylation reaction with monocyclic benzyl alcohol were investigated (Table 2 and Supplementary Table 14). Inspired by Trost's work[29,30], we chose benzyl diethyl phosphate **2b** as a benzylation reagent. As expected, this reaction proceeded smoothly giving desired product **7a** in 26% yield and 75% ee (Table 2, entry 1). The introduction of chiral ligand **L1** improved the yield of **7a** greatly (Table 2, entry 2). Then, the matching relationship between chiral aldehyde and ligand was investigated. Results indicated that the combination of chiral aldehyde **3f** and ligand (R)-L1 was suitable for this reaction (Table 2,

entry 3 *vs* entry 4). After we replaced the reactant **1a** with **1b**, product **7b** was generated in 83% yield and 95% ee (Table 2, entry 5). For the reaction of **1b** and **2b**, the usage of a single chiral catalyst, either with the chiral aldehyde **3f** or with the chiral ligand (R)-L1, could not give satisfactory experimental outcomes (Table 2, entries 6–7). According to these results, the optimal reaction conditions for the catalytic asymmetric α−benzylation of N-unprotected amino acids with monocyclic benzyl alcohols were determined.

**Substrate scopes with monocyclic benzyl alcohol derivatives and amino acid esters**. With these optimal reaction conditions

**Table 2 Reaction optimization of monocyclic benzyl alcohol involved benzylation reaction[a].**

| Entry | Ar*CHO | R | Ligand | Time (h) | yield (%)[b] | ee (%)[c] |
|---|---|---|---|---|---|---|
| 1 | **3f** | Et | dppp | 12 | 26 | 75 |
| 2[d] | **3f** | Et | (R)-**L1** | 9 | 60 | 78 |
| 3[d] | **3a** | Et | (S)-**L1** | 12 | 51 | 19 |
| 4[d] | **3a** | Et | (R)-**L1** | 12 | 60 | 72 |
| 5 | **3f** | tBu | (R)-**L1** | 9.5 | 83 | 95 |
| 6[d] | **3f** | tBu | dppp | 9.5 | 50 | 83 |
| 7 | rac-**3a** | tBu | (R)-**L1** | 9.5 | 62 | 74 |

[a]Reaction conditions: **1** (0.30 mmol), **2a** (0.20 mmol), **3** (0.02 mmol), **L1** (0.02 mmol), [Pd(C₃H₅)Cl]₂ (0.01 mmol), TDMAIP (0.28 mmol), and ZnCl₂ (0.08 mmol) were stirred in mesitylene (0.5 mL) at 60 °C for indicated time.
[b]Isolated yield.
[c]The enantioselective excess was determined by chiral HPLC.
[d]With 10 mol% palladium and 20 mol% (R)-**L1**.

listed in Table 2, entry 5, corresponding substrate scopes were investigated. Results indicated that various monocyclic benzyl phosphates bearing an ortho-, meta- or para-substituent were good acceptors for this reaction (Fig. 4, **7c–7k**). Benzyl phosphates having two substituents on the phenyl ring also participated in this reaction well (Fig. 4, **7l–7m**). Thiophen-3-ylmethanol-derived phosphate could give product **7n** in 80% yield and 96% ee. Then, five amino acids derived tert-butyl esters were tested. All of them gave corresponding products in high yields and excellent enantioselectivities (Fig. 4, **7o–7s**). Thus, both of poly- and monocyclic benzyl alcohol derivatives were successfully introduced as benzylation regents.

**Formal synthesis of SRIF mimetics.** Compounds **11a** and **11b** are somatotropin release inhibiting factor (SRIF) mimetics with IC₅₀ values of 2.44 and 1.27 μM, respectively, for the somatostatin receptor hsst 5[51]. Chiral amino acid ester **10**, which bears an aldehyde group, is one of the key chiral building blocks for the synthesis of these two SRIF mimetics. Ten steps are involved in the reported preparation of **10** from the corresponding amino acids. We envisioned that this chiral synthon could be obtained from amino acid ester **8** by sequential protection and oxidative cleavage. We chose amino acid ester **1b** as the donor in reactions with naphthalenemethanol derivatives **2a** and **2b** under the optimal reaction conditions. As expected, products **8a** and **8b** were generated in good yields and with good enantioselectivities. After protection of the amino group in **7** with Cbz, oxidative cleavage of C=C bonds was performed. The chiral amino acid esters **9a** and **9b** were obtained in three steps and can be used for the formal synthesis of SRIF mimetics **10a** and **10b** (Fig. 5a).

**Formal synthesis of hypoestestatin 1.** 13a-Methylphenanthro-indolizidine alkaloids are important members of the phenan-throindolizidine alkaloid family[52–56]. These alkaloids show a wide range of biological activities. Among them, four compounds, one of which is hypoestestatin 1, have the same core phenan-throindolizidine skeleton; their structures differ in terms of the substituents on the phenanthrene ring. Wang's work indicated that hypoestestatin 1 can be prepared from the chiral α-

phenanthryl amino acid ester **13**[57,58]. We envisioned that this chiral synthon could be obtained from γ-lactam **12** by selective reduction. The result obtained for products **5q** show that **12** can be generated by the asymmetric benzylation reaction of dimethyl glutamate and a carbonate derived from a substituted phenan-thrylmethanol. On the basis of this retrosynthetic analysis, we began to explore new synthetic routes to the proposed molecule structure of hypoestestatin 1. Under the optimal reaction conditions, **1c** and **2c** reacted smoothly to give γ-lactam **12** in 65% yield and with 86% ee. Selective reduction of the amide group of **12** gave the desired chiral synthon **13** in 62% yield and with 85% ee. According to a reported procedure, hypoestestatin 1 can be synthesized from **13** in four steps (Fig. 5b)[57]. The target natural product could therefore be obtained via six steps in total. This strategy could be potentially used to synthesize other three natural products, as shown in Fig. 5b.

**Discussion**

The possible reaction mechanism was then investigated. Generally, the chiral aldehyde catalyst generates an active enolate via sequential Schiff base formation and deprotonation, and the palladium catalyst generates an active electrophile by oxidative addition. The key issue is the identification of the transition state involved in this reaction. This was clarified by performing various control experiments (Fig. 6a). Under the optimal reaction conditions, chiral aldehyde **3a** gave product **5a** in 85% yield and with 73% ee. In the absence of the Lewis acid ZnCl₂, both the yield and enantioselectivity of **5a** decreased. This indicates the possible formation of a Zn²⁺–Schiff base complex during the reaction. Two modified chiral aldehydes were then used as catalysts. We found that chiral aldehyde **3o** did not promote this reaction efficiently. Chiral aldehyde **3p** gave product **5a** in 80% yield, but the ee was only 27%. The results of these two control experiments indicate that the 2'-hydroxyl group in chiral aldehydes **3** is crucial for this reaction. We envisioned that the 2'- hydroxyl group in the chiral aldehyde catalyst acts as a site for coordination with an active η³-benzyl–palladium complex (E⁺). The linear relationship between the ee values of **3f** and those of the product **5a** indicated that one molecule of chiral aldehyde catalyst participated in the stereoselective control model. In addition, we found the Schiff

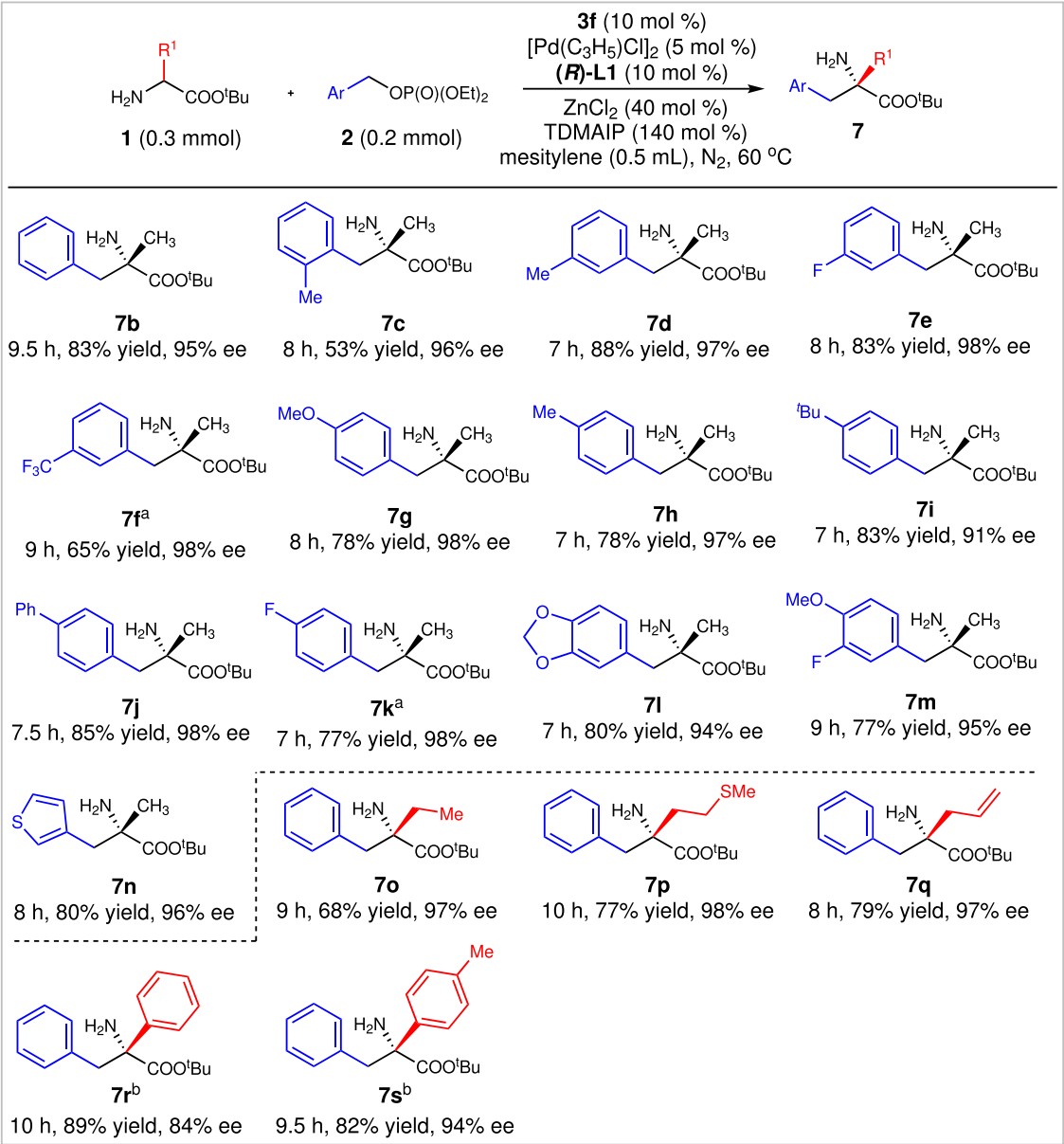

**Fig. 4 Substrate scope of monocyclic benzyl alcohol derivatives and amino acid esters.** Reaction conditions: **1** (0.30 mmol), **2** (0.20 mmol), **3f** (0.02 mmol), (R)-**L1** (0.02 mmol), [Pd(C₃H₅)Cl]₂ (0.01 mmol), tris(dimethylamino)iminophosphorane (TDMAIP, 0.28 mmol), and ZnCl₂ (0.08 mmol) were stired in mesitylene (0.5 mL) at 60 °C under the nitrogen atmosphere for indicated time. Isolated yields. The enantioselective excess (ee) was determined by HPLC. [a]Ee values were given by the N-Cbz-protected derivatives. [b]With 10 mol% palladium, 20 mol% (R)-**L1**, and at 80 °C.

base **I** was an efficient catalyst and produced **5a** in 90% yield and 89% ee, which indicated a possible amine exchange process existed in the catalytic cycle. Based on these results, a transition state **TS** and possible catalytic cycles were proposed (Fig. 6b). In this transition state, the Si face of the enolate undergoes an intramolecular electrophilic attack to give Schiff base **V** enantioselectively. Finally, the chiral aldehyde catalytic cycle is completed by hydrolysis or amine exchange and the palladium catalytic cycle is completed by reductive elimination (Fig. 6b).

The key intermediates involved in this proposed reaction mechanism were then verified by HRMS with negative ion mode (Fig. 6c and Supplementary Figs. 2–6). The Schiff bases **I** ($m/z = 484.1966$, M–H) and **VII** ($m/z = 624.2575$, M–H) could be observed directly in the reaction system. Although we could not find the key intermediates **II** and **V** directly, three fragments derived from them were observed and confirmed by comparing

their isotopic distribution with theoretical data. With the dissociation of ethyl, ionic fragments **III** ($m/z = 554.0551$, M) and **VI** ($m/z = 694.1142$, M) were generated from intermediates **II** and **V**, respectively. It was because the intermediates **II** and **V** were not stable enough for HRMS detection; with the activation of Lewis acid ZnCl₂, the C–O bond of the ethoxy was readily dissociated by the electron impact of ESI source. As result, stable ionic fragments **III** and **VI** were generated. A neutral intermediate **IV** ($m/z = 518.0749$, M–H), which was most likely generated from fragment **III** with the dissociation of a Cl⁻, was also detected by HRMS. All of the isotopic distributions of **III**, **IV**, and **VI** were following the theoretical data (Supplementary Figs. 3–5). The existence of fragments **III**, **IV**, and **VI** confirmed the formation of Schiff base—Zn complexes **II** and **V** in this reaction.

In this work, we have developed a highly efficient asymmetric α-benzylation reaction of N-unprotected amino acids and

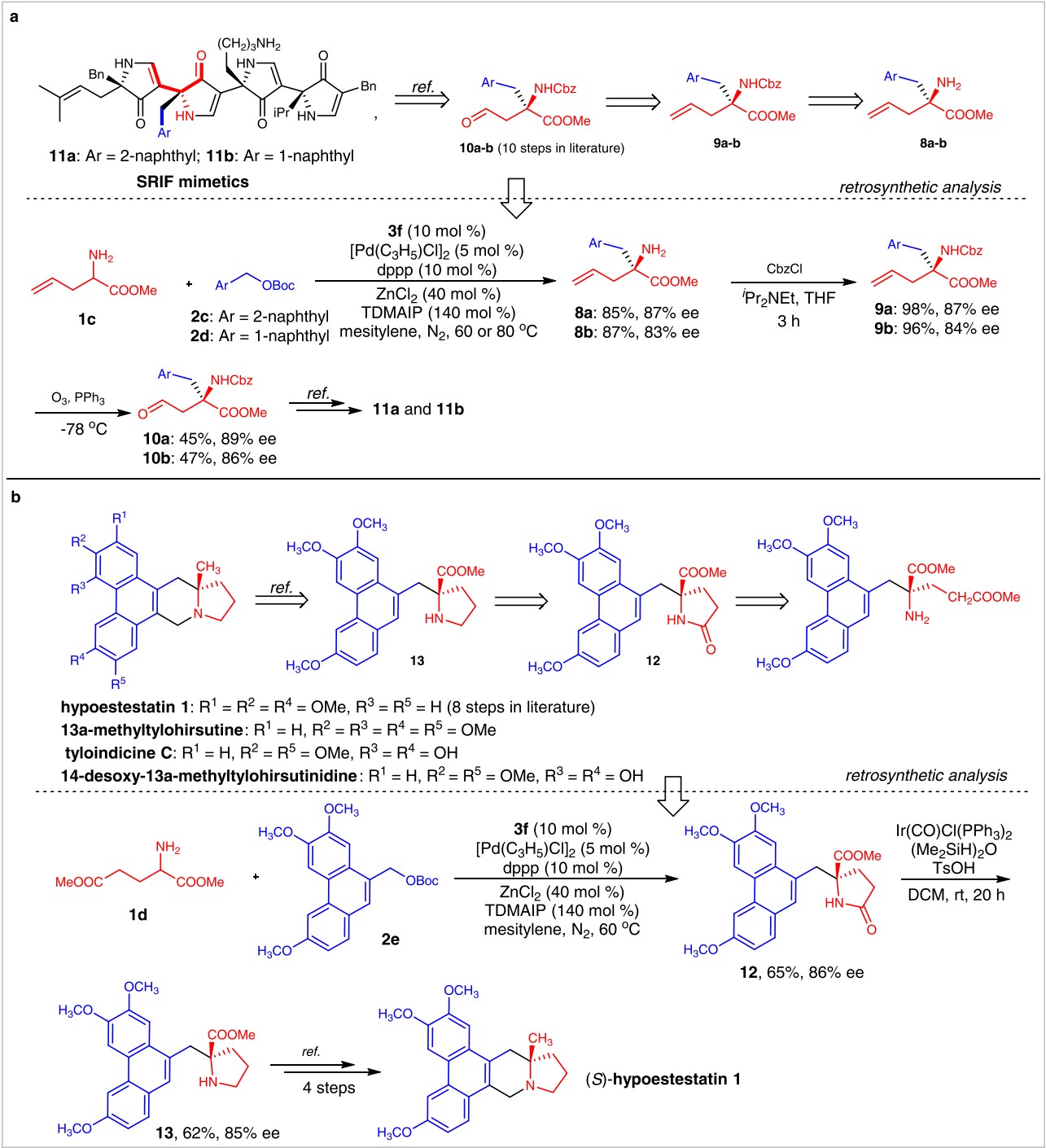

**Fig. 5 Synthetic applications. a** The retrosynthetic analysis and formal synthesis of somatotropin release inhibiting factor (SRIF) mimetics. **b** The retrosynthetic analysis and formal synthesis of (*S*)-hypoestestatin **1**. TDMAIP   tris(dimethylamino)iminophosphorane.

arylmethanols. The reaction is promoted by a combination of catalytic systems, namely a chiral aldehyde, a palladium species, and a Lewis acid. Various chiral α-benzyl amino acids are produced in high-to-excellent yields and with high-to-excellent enantioselectivities. This strategy can be conveniently used in the preparation of SRIF mimetics and the formal synthesis of the natural product (*S*)-hypoestestatin 1. Based on the results of control experiments and nonlinear effect investigation, a reasonable reaction mechanism is proposed. The key intermediates involved in the chiral aldehyde catalytic cycle are confirmed by HRMS detection.

## Methods

### General procedure for the catalytic asymmetric α−benzylation of amino acids.
To a 10 mL vial charged with [Pd(C₃H₅)Cl]₂ (3.6 mg, 0.01 mmol) and ligand (dppp or *R*-**L1**) (0.02 mmol) was added 0.5 mL mesitylene, and the mixture was stirred under nitrogen atmosphere at room temperature for 30 min. Then, ethyl amino acid ester **1** (0.3 mmol), benzyl alcohol derivative **2** (0.2 mmol), chiral aldehyde **3f** (7.7 mg, 0.02 mmol), ZnCl₂ (10.9 mg, 0.08 mmol) and TDMAIP (50.9 μL, 0.28 mmol) were added. The mixture was continuously stirred at indicated reaction temperature under a nitrogen atmosphere. After the reaction completed, the solvent was removed by rotary evaporation, and the residue was purified by flash chromatography column on silica gel (eluent: petroleum ether/ ethyl acetate/triethylamine = 200/100/3). The details of the full experiments and compound characterizations are provided in the Supplementary Information.

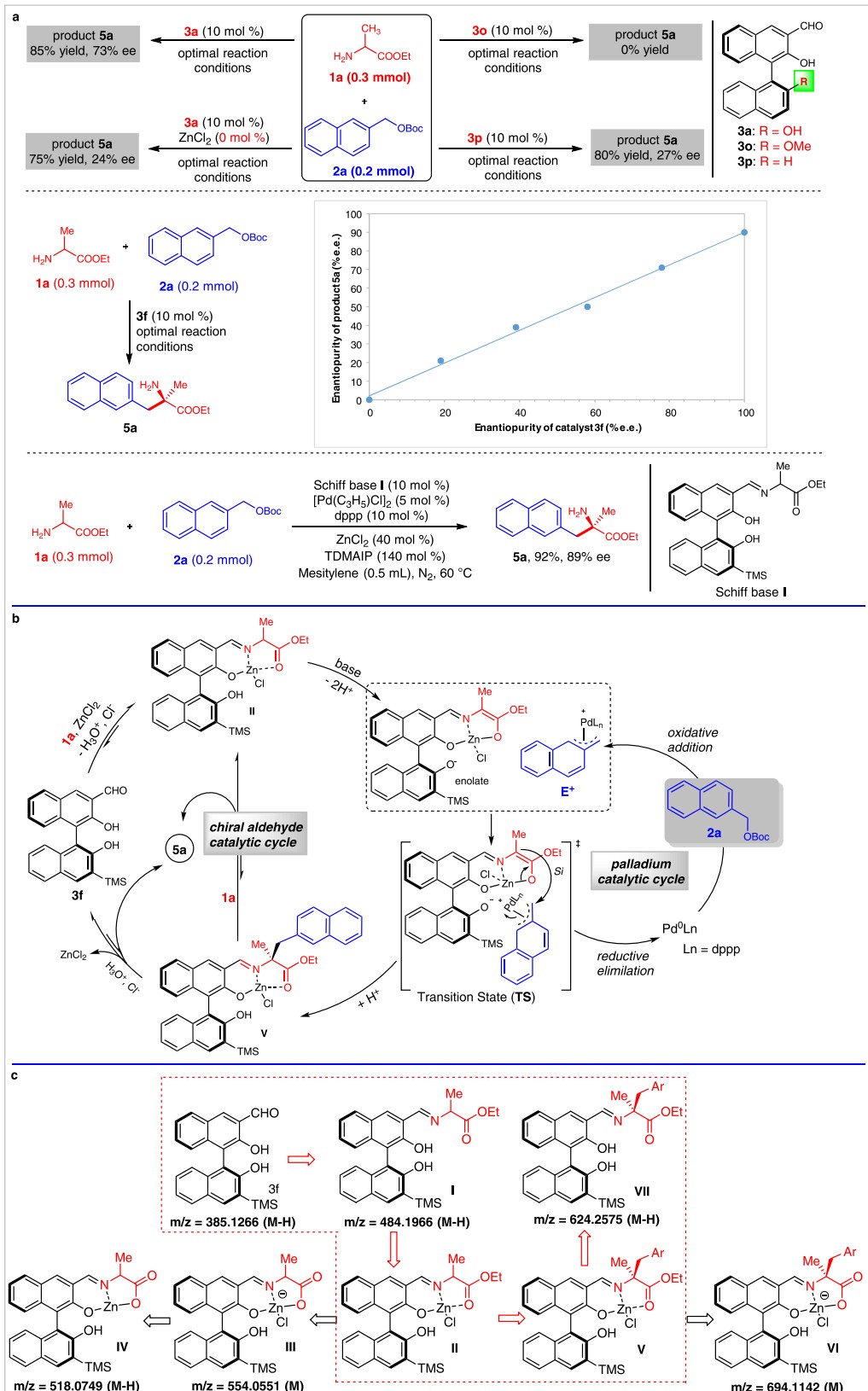

**Fig. 6 Reaction mechanism investigations. a** Control experiments and nonlinear effect investigation. **b** Proposed catalytic cycles. **c** Key intermediated detected by HRMS. TDMAIP tris(dimethylamino)iminophosphorane.

## Data availability
The authors declare that all other data supporting the findings of this study are available within the article and its Supplementary Information file.

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

## Acknowledgements

We are grateful for financial support from NSFC (22071199, 21871223) and the Chongqing Science Technology Commission (cstccxljrc201701, cstc2018jcyjAX0548).

## Author contributions

W.W. and G.Q.X. conceived this project. L.J.H., L.J., S.Q.W., and L.Y. carried out the experiments. W.Z.L. and C.T. performed the HRMS analysis. G.Q.X. wrote the manuscript. All authors discussed the results.

## Competing interests

The authors declare no competing interests.
