## [Peer Review File · Nature Communications]

REVIEWER COMMENTS

Reviewer #1 (Remarks to the Author):

The manuscript by Qi-Xiang Guo and coworkers describes the enantioselective benzylation of unprotected amino acids catalyzed by the catalytic system composed of a palladium complex, zinc chloride, and a chiral aldehyde. This process can be readily applied to the catalytic reactions between a range of amino acids and naphthyl/phenyl methanol derivatives, thereby generating chiral benzylic-amino acids in good yields with high enantioselectivity. This manuscript also shows the application of this method for the synthesis of complex molecules, somaostatin mimetics and hypoestestatin 1. While this work introduces highly useful benzylation reaction, it seems to be the extension of the previous work on the corresponding allylation reported by the author's group (refs 35-36) which employs the nearly identical, ternary catalytic system; not only the reaction conditions but also the mechanistic discussions of this manuscript are similar to those of the previous reports---no new mechanistic insights, other than the participation of a Pd- π -benzyl complex instead of a Pd- π -allyl complex. Therefore, unfortunately, this reviewer has concluded that the conceptual novelty of this work is not sufficient to justify the publication in Nature Communications.

Reviewer #2 (Remarks to the Author):

The manuscript entitled "Catalytic Asymmetric Tsuji-Trost α -Benzylation Reaction of N-Unprotected Amino Acids and Benzyl alcohols" discloses a Pd/aldehyde-cocatalyzed enantioselective benzylation of N-unprotected α -amino acid derivatives. As the authors mentioned, the Pd-catalyzed asymmetric benzylation via π -benzyl Pd intermediate has been less developed, despite its isoelectronic character to the well-established π -allyl Pd intermediate. The authors use a catalytic amount of the rationally designed BINOL-derived aldehyde as the chiral promotor to successfully develop the otherwise challenging asymmetric benzylation of N-unprotected α -amino acids. Additionally, in the case with less reactive monocyclic benzyl electrophiles, the chiral Pd/chiral phosphine/chiral aldehyde ternary system works well to deliver the targeted benzylated products in good yields with high enantioselectivity. The mechanistic studies seems to be still somewhat premature, but the proposed reaction mechanism is reasonably supported by some control experiments. Thus, this reviewer believes that this work can greatly contribute to further development of asymmetric catalysis and stereoselective organic synthesis.

On the basis of the above considerations, I can support acceptance in Nature Commun. Only the following minor points should be reconsidered before final acceptance.

- 1) In the manuscript title, "benzyl alcohols" should be changed to "benzyl alcohol derivatives".
- 2) The authors should mention and cite the following important contributions to develop the Pd-catalyzed asymmetric benzylation: OL 2017, 19, 2438 (N- and O-benylation); CEJ 2018, 24, 6525 (S-benylation); OL 2018, 20, 3553 (P-benylation); Sci. China Chem. 2020, 63, 687 (C-benylation by Pd/PC dual catalyst); OL 2022, 24, 1258 (P-benylation under related asymmetric Ni catalysis).
- 3) In Table 4, the choice of ester moiety greatly affected the enantioselectivity (entry 5). Thus, the authors should show the result of t-Bu ester 1b with 2b under the combination of achiral Pd/dppb and chiral aldehyde 3f in the main manuscript rather than the Supporting Information. Additionally, the control experiment with (R)-L1 and racemic 3a or 3f should be performed.
- 4) In entry 2 of Table 4, dppp in the column "L" should be changed to (R)-L1.
- 5) In Table 5 and its caption, L1 should be corrected to (R)-L1.
- 6) As the mechanistic consideration, the authors should monitor the formation of the Schiff base intermediate I and enolate by using ¹H NMR.

Reviewer #3 (Remarks to the Author):

Guo and coworkers described a highly enantioselective α -benzylation reaction of N-protected amino acids employing an efficient ternary catalyst system consisting of a chiral aldehyde, a palladium complex, and a Lewis acid. This is a rather difficult transformation that has witnessed much less successes compared to the standard asymmetric Tsuji-Trost allylation reactions. Highly active and efficient catalytic systems are required to overcome the energy barrier of the dearomatization process under mild conditions to meet the requirement of enantioselectivity control at the same time.

A rather broad substrate scope is presented, including naphthyl methanol and heteroaryl methanol derivatives, as well as simple benzylic alcohol esters, which also provided excellent enantioselectivities. The utility-demonstration also seems quite convincing.

Overall, I feel confident to recommend this work for publication in Nat. Commun.

Typos and suggestions:

Since the reaction employed benzyl alcohol esters as the substrates, it would be more appropriate to change "Benzyl alcohols" to "Benzyl Alcohol Derivatives" or "Benzyl Alcohol Esters" in the title, as well as some other parts in the manuscript.

Response to Referees

Dear referees,

Thanks for your valuable and expert comments on our work very much. We carefully revised our manuscript and provided our revisions and responses to your comments by point-to-point in this letter.

1. Responses and revisions to the comments of Reviewer 1

Comments: The manuscript by Qi-Xiang Guo and coworkers describes the enantioselective benzylation of unprotected amino acids catalyzed by the catalytic system composed of a palladium complex, zinc chloride, and a chiral aldehyde. This process can be readily applied to the catalytic reactions between a range of amino acids and naphthyl/phenyl methanol derivatives, thereby generating chiral benzylic-amino acids in good yields with high enantioselectivity. This manuscript also shows the application of this method for the synthesis of complex molecules, somaostatin mimetics and hypoestestatin 1. While this work introduces highly useful benzylation reaction, it seems to be the extension of the previous work on the corresponding allylation reported by the author's group (refs 35-36) which employs the nearly identical, ternary catalytic system; not only the reaction conditions but also the mechanistic discussions of this manuscript are similar to those of the previous reports--- no new mechanistic insights, other than the participation of a Pd- π -benzyl complex instead of a Pd- π -allyl complex. Therefore, unfortunately, this reviewer has concluded that the conceptual novelty of this work is not sufficient to justify the publication in Nature Communications.

Our response and revisions: Thanks for your comments very much. Although the ternary catalytic system is similar to that we reported previously, undoubtedly, this strategy provides an excellent solution for the challenging asymmetric α -benzylation of *N*-unprotected amino acids with aryl methanol derivatives and provides a straightforward method leading to market valuable chiral α -benzyl amino acids. Furthermore, good synthetic applications are also presented. In this revision, we gave much more studies on the reaction mechanism. Especially, we carried out a control experiment to verify the last step of the chiral aldehyde catalytic cycle, took a nonlinear effect investigation to explore the possible transition state, and detected the key intermediates involved in this reaction by HRMS. Based on these studies, the catalytic cycles were further fleshed out. Details of our revisions on the mechanism investigation please see Figure 2 in

Revised Manuscript and Figures S1-6 in Revised Supplementary Information.

2. Revisions to the comments of Reviewer 2

Comments: The manuscript entitled “Catalytic Asymmetric Tsuji–Trost α -Benzylation Reaction of N-Unprotected Amino Acids and Benzyl alcohols” discloses a Pd/aldehyde-cocatalyzed enantioselective benzylation of N-unprotected α -amino acid derivatives. As the authors mentioned, the Pd-catalyzed asymmetric benzylation via π -benzyl Pd intermediate has been less developed, despite its isoelectronic character to the well-established π -allyl Pd intermediate. The authors use a catalytic amount of the rationally designed BINOL-derived aldehyde as the chiral promotor to successfully develop the otherwise challenging asymmetric benzylation of N-unprotected α -amino acids. Additionally, in the case with less reactive monocyclic benzyl electrophiles, the chiral Pd/chiral phosphine/chiral aldehyde ternary system works well to deliver the targeted benzylated products in good yields with high enantioselectivity. The mechanistic studies seems to be still somewhat premature, but the proposed reaction mechanism is reasonably supported by some control experiments. Thus, this reviewer believes that this work can greatly contribute to further development of asymmetric catalysis and stereoselective organic synthesis.

On the basis of the above considerations, I can support acceptance in Nature Commun. Only the following minor points should be reconsidered before final acceptance.

Question 1: In the manuscript title, “benzyl alcohols” should be changed to “benzyl alcohol derivatives”.

Our revision: Thanks for your reminder very much. We corrected the “benzyl alcohols” to “benzyl alcohol derivatives” in our Revised Manuscript.

Question 2: The authors should mention and cite the following important contributions to develop the Pd-catalyzed asymmetric benzylation: OL 2017, 19, 2438 (N- and O-benzylation); CEJ 2018, 24, 6525 (S-benzylation); OL 2018, 20, 3553 (P-benzylation); Sci. China Chem. 2020, 63, 687 (C-benzylation by Pd/PC dual catalyst); OL 2022, 24, 1258 (P-benzylation under related asymmetric Ni catalysis).

Our revision: Thanks for your suggestion very much. All of the mentioned references were cited. Details please see the refs. 23-26 and 33 in Revised Manuscript. Other reference numbers were tuned accordingly.

Question 3: In Table 4, the choice of ester moiety greatly affected the enantioselectivity (entry

5). Thus, the authors should show the result of t-Bu ester 1b with 2b under the combination of achiral Pd/dppb and chiral aldehyde 3f in the main manuscript rather than the Supporting Information. Additionally, the control experiment with (R)-L1 and racemic 3a or 3f should be performed.

Our revision: Thanks for your suggestion very much. Results of these two control experiments were added in Table 5 of the Revised manuscript.

Question 4: In entry 2 of Table 4, dppp in the column "L" should be changed to (R)-L1.

Our revision: Thanks for your reminder very much. The wrongly inputted ligand name and experimental results in Table 4, entry 2, were corrected.

Question 5: In Table 5 and its caption, L1 should be corrected to (R)-L1.

Our revision: Thanks for your reminder very much. The wrongly imputed L1 was corrected as (R)-L1.

Question 6: As the mechanistic consideration, the authors should monitor the formation of the Schiff base intermediate I and enolate by using ¹H NMR.

Our response and revisions: Thanks for your comments and suggestion very much. We tried our best to monitor the key intermediates involved in this reaction by ¹H NMR, but failed. We found there were two difficulties that hindered us from successfully trapping the key intermediates by ¹H NMR. One is the lack of standard samples. According to some reported literatures, the Schiff base formed from salicylaldehyde analogue and amino acid can coordinate with Zn²⁺ and produce a stable Schiff base-Zn complex; however, once the amino acid was replaced by an amino acid ester, the formed Schiff base could not produce stable Schiff base-Zn complex. So, only two Schiff bases, one is formed from chiral aldehyde and amino acid ester reactant, the other is formed from chiral aldehyde and amino acid ester product, was successfully prepared as standard samples. During we monitored the reaction system by ¹H NMR, we found the chemical shift of the Schiff bases changed, but we could not assure if the Schiff base-Zn complex was formed in situ. The other difficulty is that our reaction system is too complicated. There are too many ¹H NMR peaks that could interfere with our detection.

HRMS is another efficient approach for reaction mechanism investigation, especially for the trapping of active intermediates. We adopted HRMS to monitor our reaction system. We found two Schiff bases and three fragments of Schiff base-Zn complexes were observed directly, which supported our proposed reaction mechanism well. Through this HRMS detection, we also found

the chlorine anion participated in the transition states, which is a new discovery. We corrected our reaction mechanism accordingly. Additionally, the result of a nonlinear effect investigation was provided. Details of our revisions about the reaction mechanism investigation please see Figure 2 in Revised Manuscript and Figures S1-6 in Revised Supplementary Information.

3. Revisions to Reviewer 3

Comments: Guo and coworkers described a highly enantioselective alfa-benylation reaction of N-protected amino acids employing an efficient ternary catalyst system consisting of a chiral aldehyde, a palladium complex, and a Lewis acid. This is a rather difficult transformation that has witnessed much less successes compared to the standard asymmetric Tsuji-Trost allylation reactions. Highly active and efficient catalytic systems are required to overcome the energy barrier of the dearomatization process under mild conditions to meet the requirement of enantioselectivity control at the same time.

A rather broad substrate scope is presented, including naphthyl methanol and heteroaryl methanol derivatives, as well as simple benzylic alcohol esters, which also provided excellent enantioselectivities. The utility-demonstration also seems quite convincing.

Overall, I feel confident to recommend this work for publication in Nat. Commun.

Typos and suggestions:

Since the reaction employed benzyl alcohol esters as the substrates, it would be more appropriate to change “Benzyl alcohols” to “Benzyl Alcohol Derivatives” or “Benzyl Alcohol Esters” in the title, as well as some other parts in the manuscript.

Our response and revisions: Thanks for your suggestion very much. We corrected the ‘benzyl alcohols’ to ‘benzyl alcohol derivatives’ in the Revised Manuscript.

All of the above revisions were marked as the red font in our Revised Manuscript.

Hopefully, this revised version of our manuscript can meet the high requirement of Nature Communications.

Sincerely yours,

Best wishes,

Dr. Guo

REVIEWERS' COMMENTS

Reviewer #2 (Remarks to the Author):

I have reviewed the original submission of this manuscript. The authors carefully revised it according to comments and suggestions by all reviewers. Unfortunately, the solid evidence of the proposed mechanism cannot be obtained, but additional investigation using HRMS provides some new important mechanistic aspects. Thus, I believe that the revised version meets the criteria of novelty and urgency for publication in Nat. Commun. Additional scientific revision and review process are not needed.

Just a comment:

Very recently, a highly related Pd-catalyzed asymmetric benzylation of alpha-amino acid derivatives appeared in Or. Lett., which should be mentioned and cited (doi: 10.1021/acs.orglett.2c00865).

Reviewer #3 (Remarks to the Author):

As my comments gave in the first peer review, this work provided an excellent solution for achieving the challenging catalytic asymmetric alpha-benylation reaction. The unique property of the chiral aldehyde-involved catalytic system allowed the direct usage of N-unprotected amino acid esters as reactants, and this combined catalytic system exhibited powerful activation and excellent stereoselective control abilities. Especially, both mono- and polycyclic benzyl alcohols were good reaction partners and could produce corresponding products with excellent yields and enantioselectivities. In this revision, additional reaction mechanism investigations were provided. Especially, the detection of the key intermediates that possibly involved in the reaction by HRMS furtherly fleshed out the immature study in the first version. Based on these reasons, I think that the quality of this manuscript by Guo et al. can meet the extremely high requirement of Nat. Commun., and strongly recommend it to be accepted and published in this journal.

Response to Referees

Dear referees,

Thanks for your valuable and expert comments on our work very much. We carefully revised our manuscript and provided our revisions and responses to your comments by point-to-point in this letter.

1. Responses and revisions to the comments of Reviewer 2

Comments: I have reviewed the original submission of this manuscript. The authors carefully revised it according to comments and suggestions by all reviewers. Unfortunately, the solid evidence of the proposed mechanism cannot be obtained, but additional investigation using HRMS provides some new important mechanistic aspects. Thus, I believe that the revised version meets the criteria of novelty and urgency for publication in *Nat. Commun.* Additional scientific revision and review process are not needed.

Just a comment:

Very recently, a highly related Pd-catalyzed asymmetric benzylation of alpha-amino acid derivatives appeared in *Org. Lett.*, which should be mentioned and cited (doi: 10.1021/acs.orglett.2c00865).

Our response and revisions: Thanks for your comments very much. During the review process of our manuscript, two works about the catalytic asymmetric α -benzylation of N-protected amino acid esters were published. We added them as *ref. 34* (*Angew. Chem. Int. Ed.* **2022**, DOI: 10.1002/anie.202203448) and *ref. 35* (*Org. Lett.* **2022**, 24, 2573-2578) in our revised manuscript. Other references' numbers were adjusted accordingly.

2. Response to the comments of Reviewer 3

Comments: As my comments gave in the first peer review, this work provided an excellent solution for achieving the challenging catalytic asymmetric α -benzylation reaction. The unique property of the chiral aldehyde-involved catalytic system allowed the direct usage of N-unprotected amino acid esters as reactants, and this combined catalytic system exhibited powerful activation and excellent stereoselective control abilities. Especially, both mono- and polycyclic benzyl alcohols were good reaction partners and could produce corresponding products with excellent yields and enantioselectivities. In this revision, additional reaction

mechanism investigations were provided. Especially, the detection of the key intermediates that possibly involved in the reaction by HRMS furtherly fleshed out the immature study in the first version. Based on these reasons, I think that the quality of this manuscript by Guo et al. can meet the extremely high requirement of Nat. Commun., and strongly recommend it to be accepted and published in this journal.

Our response: Thank you very much. It's our honor to receive your positive comments.

All of the above revisions were marked as the red font in our Revised Manuscript.

Hopefully, this revised version of our manuscript can meet the high requirement of Nature Communications.

Sincerely yours,

Best wishes,

Dr. Guo